# The Role of Molossidae and Vespertilionidae in Shaping the Diversity of Alphacoronaviruses in the Americas

Diego A. Caraballo,[a,b] María S. Sabio,[c] Valeria C. Colombo,[c,d] María Guadalupe Piccirilli,[c] Lorena Vico,[g] Stella Maris Hirmas Riade,[c] Josefina Campos,[e] Gustavo Martínez,[g] Fernando Beltrán,[h] Elsa Baumeister,[f] Daniel M. Cisterna[c]

[a]CONICET-Universidad de Buenos Aires, Instituto de Ecología, Genética y Evolución de Buenos Aires (IEGEBA), Ciudad Universitaria-Pabellón II, Ciudad Autónoma de Buenos Aires, Argentina

[b]Universidad de Buenos Aires, Facultad de Ciencias Exactas y Naturales, Buenos Aires, Argentina

[c]Servicio de Neurovirosis, Instituto Nacional de Enfermedades Infecciosas, Administración Nacional de Laboratorios e Institutos de Salud (ANLIS), Dr. Carlos G. Malbrán, Ciudad Autónoma de Buenos Aires, Argentina

[d]Evolutionary Ecology Group, Department of Biology, University of Antwerp, Antwerp, Belgium

[e]Unidad de Genómica y Bioinformática, Administración Nacional de Laboratorios e Institutos de Salud (ANLIS), Dr. Carlos G. Malbrán, Ciudad Autónoma de Buenos Aires, Argentina

[f]Servicio de Virosis Respiratorias, Instituto Nacional de Enfermedades Infecciosas, Administración Nacional de Laboratorios e Institutos de Salud (ANLIS), Dr. Carlos G. Malbrán, Ciudad Autónoma de Buenos Aires, Argentina

[g]Departamento de Zoonosis Urbanas, Avellaneda, Provincia de Buenos Aires, Argentina

[h]Instituto de Zoonosis Dr. Luis Pasteur, Ciudad Autónoma de Buenos Aires, Argentina

**ABSTRACT** Bats are reservoirs of diverse coronaviruses (CoVs), including progenitors of severe acute respiratory syndrome CoV (SARS-CoV) and SARS-CoV-2. In the Americas, there is a contrast between alphacoronaviruses (alphaCoVs) and betaCoVs: while cospeciation prevails in the latter, alphaCoV evolution is dominated by deep and recent host switches. AlphaCoV lineages are maintained by two different bat family groups, Phyllostomidae and Vespertilionidae plus Molossidae. In this study, we used a Bayesian framework to analyze the process of diversification of the lineages maintained by Molossidae and Vespertilionidae, adding novel CoV sequences from Argentina. We provide evidence that the observed CoV diversity in these two bat families is shaped by their geographic distribution and that CoVs exhibit clustering at the level of bat genera. We discuss the causes of the cocirculation of two independent clades in *Molossus* and *Tadarida* as well as the role of *Myotis* as the ancestral host and a major evolutionary reservoir of alphaCoVs across the continent. Although more CoV sampling efforts are needed, these findings contribute to a better knowledge of the diversity of alphaCoVs and the links between bat host species.

**IMPORTANCE** Bats harbor the largest diversity of coronaviruses among mammals. In the Americas, seven alphacoronavirus lineages circulate among bats. Three of these lineages are shared by members of two bat families: Vespertilionidae and Molossidae. Uncovering the relationships between these coronaviruses can help us to understand patterns of cross-species transmission and, ultimately, which hosts are more likely to be involved in spillover events. We found that two different lineages cocirculate among the bat genera *Molossus* and *Tadarida*, which share roosts and have common viral variants. The bat genus *Myotis* functions as a reservoir of coronavirus diversity and, as such, is a key host. Although there were some spillovers recorded, there is a strong host association, showing that once a successful host jump takes place, it is transmitted onward to members of the same bat genus.

**KEYWORDS** virus, coronavirus, cross-species transmission, spillover, host shift, bats, phylogeny, Molossidae, Vespertilionidae

Address correspondence to Diego A. Caraballo, dcaraballo@ege.fcen.uba.ar, or Daniel M. Cisterna, dcisterna@anlis.gob.ar.

The authors declare no conflict of interest.

Coronaviruses (CoVs), single-stranded positive-sense RNA viruses with the largest nonsegmented RNA viral genomes (16 to 31 kb), belong to the *Coronaviridae* family and the *Orthocoronavirinae* subfamily and are divided into four genera: *Alphacoronavirus* (alphaCoV), *Betacoronavirus* (betaCoV), *Gammacoronavirus* (gammaCoV), and *Deltacoronavirus* (deltaCoV). Alpha- and betaCoVs are found in mammals, while the latter two are found mainly in birds (1). By virtue of their large genome size, high recombination rates, and genomic plasticity, CoVs are able to jump cross-species barriers and rapidly adapt to new hosts (2, 3).

Bats harbor the largest diversity of CoVs among mammals, with alphaCoVs being more widespread and abundant than betaCoVs (4). Recent studies suggest that bat CoV diversity is correlated with host taxonomic diversity, with the highest viral diversity being found in areas with the highest levels of bat species richness (5, 6). Bat CoVs are able to jump to different mammalian hosts, including humans, with bats being the likely ancestral hosts for all human-infective alpha- and betacoronaviruses, the latter including SARS-CoV, Middle East respiratory syndrome-related coronavirus (MERS-CoV), and severe acute respiratory syndrome coronavirus 2 (SARS-CoV-2), highlighting the relevance of bat CoVs to global health (3).

In the Americas, bat CoVs have been detected in 11 countries (7). The analysis of cross-species transmission (CST) yielded contrasting results between alpha- and betaCoVs (8). The diversification of betaCoVs accompanied the diversification of host species at the family, genus, and species levels, with no CSTs being found in the studied region. In contrast, both deep and shallow CSTs prevailed among alphaCoVs, as evidenced by the existence of multiple reciprocally monophyletic groups interspersed in the global CoV tree and by the high number of terminal CSTs between nonrelated species. However, the occurrence of related viral sequences found among related bat species (or conspecifics) in distant sampling locations is reflective of some level of codivergence between alphaCoVs and bats.

Four of the seven alphaCoV clades found in the Americas are maintained by Phyllostomidae, which was pointed out as a key bat family in the origins and diversification of alphaCoVs, based on high CST rates (8, 9). The remaining three alphaCoV clades are maintained by members of the Vespertilionidae and Molossidae, suggesting a possible link between these two bat families (8).

In this study, we explore the epidemiological links between molossid and vespertilionid bats in the Americas, also contributing the first CoV data set from Argentina. A phylogenetic Bayesian framework was applied to reconstruct ancestral host species and the spatial location of bat CoVs. The levels of geographic and host structure and the phylogenetic diversity of each host genus were also analyzed. We discuss the role of these two bat families in the origins and maintenance of CoV diversity across the continent.

## RESULTS

**Bat CoV discovery in Argentina.** From a total of 594 bats studied, 49 (8.25%) were positive for CoV using pancoronavirus real-time PCR. We obtained 21 partial sequences of 389 to 576 bp in length, spanning positions 1851 to 2426 of the RNA-dependent RNA polymerase (RdRp) coding sequence, which were used for subsequent phylogenetic analyses.

The American bat alphaCoVs were clustered into seven clades (A to G), as described in a previous study (8). The 21 sequences from Argentina pertain to clades A and B (Fig. 1) and were isolated from *Tadarida brasiliensis*, *Molossus* spp., and *Myotis* spp.

**Ancestral state reconstruction.** Although we performed ancestral state reconstructions along both clades A and B, we focus our analysis on those CSTs occurring at shallow levels of the phylogeny. The reason for excluding ancestral host shifts is that the sampling across the continent is likely incomplete (i.e., there is an obvious gap between the United States-Canada and Argentina-Brazil). In contrast, shallow host shifts are more ascertainable since these occurred in the same geographic region between sympatric hosts.

Clade A includes a sequence isolated from *Molossus* (Argentina), related to a large monophyletic group that includes Brazilian sequences from this genus, and a novel *Tadarida* lineage from Argentina (Fig. 2). The ancestral state reconstruction revealed three spillovers from *Molossus* to *Glossophaga* and *Eptesicus*.

Clade B contains alphaCoV sequences isolated from bats of the genera *Myotis*, *Tadarida*, and *Molossus* from Argentina (Fig. 3). Sample P207_20 (*Myotis*) belongs to a *Myotis* (plus one

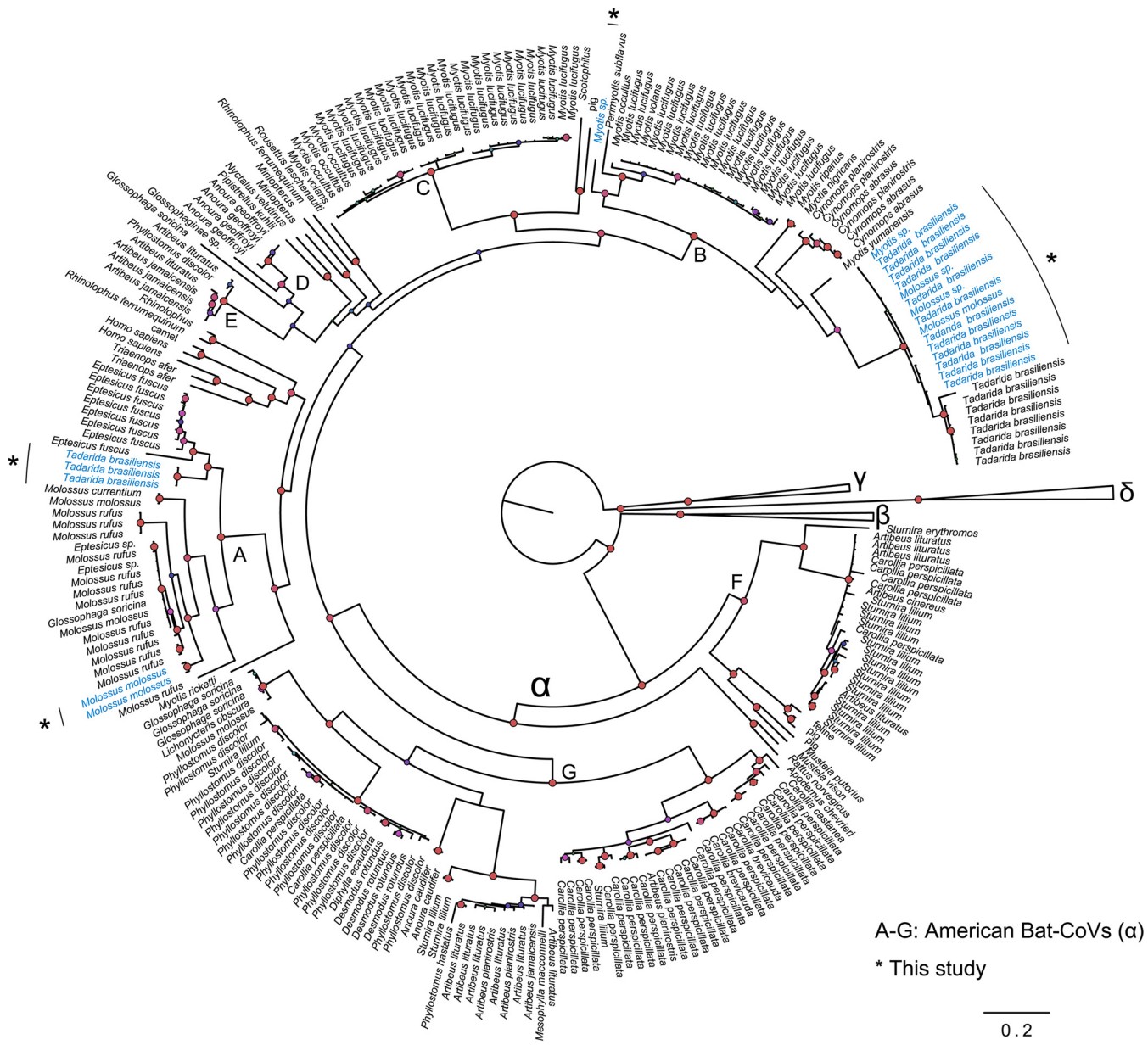

**FIG 1** American bat CoVs in the context of global CoV diversity. The Bayesian phylogeny shows global alpha-, beta-, delta-, and gammacoronaviruses. American bat alphaCoVs are shown with the letters A to G. Sequences obtained in this study are shown in blue and indicated with an asterisk. Nodes with a Bayesian posterior probability of >0.9 are shown in red. The scale bar shows substitutions per site.

*Perimyotis*) clade from Canada and the United States. An additional subclade comprising all Argentinian *Tadarida* sequences is sister to another *Tadarida* subclade involving samples from Brazil and the United States. Within the former, a *Tadarida*-to-*Myotis* spillover and a host shift from *Tadarida* to *Molossus* were identified. It is worth noting that the three samples isolated from *Molossus* form a monophyletic group within this lineage, suggesting that this variant is also maintained by mastiff bats. The ancestral state reconstruction suggests that the ancestral host for clade B is *Myotis*, which is also supported by the higher phylogenetic diversity (see below).

**Phylogenetic structure and diversity.** Tip association tests showed significant clustering for hosts and sampled countries, with none of the observed association index (AI) or parsimony score (PS) distributions overlapping the null distributions (Table 1). The monophyletic clade (MC) size statistic measures the number of tips that have a given trait value within a monophyletic group. This statistic is positively correlated with the strength of the phylogeny-trait association. In clade A, the MC was significant in *Eptesicus* and *Tadarida* but

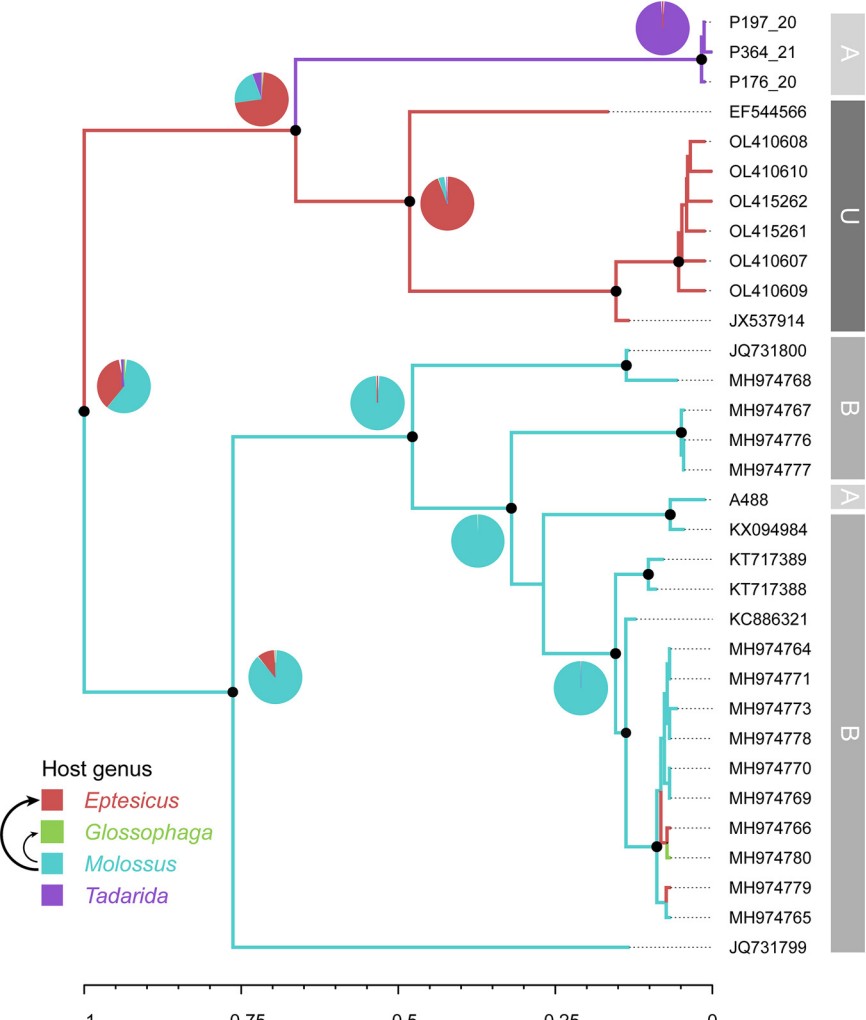

**FIG 2** Phylogenetic tree and ancestral host reconstruction of clade A. Maximum clade credibility trees were annotated using the alphaCoV data set of partial RdRp sequences and bat host genus as a discrete character state. Branch colors correspond to the inferred ancestral bat genus with the highest posterior probability. Pie charts represent uncertainty in the ancestral host reconstruction. Black arrows depict events of cross-species transmission between bat genera (with the thickness being proportional to the number of events). Branch lengths are scaled to relative time units. Relevant nodes with a posterior probability of >0.9 are indicated with black circles. Gray blocks at the right of the tree indicate the country (A, Argentina; B, Brazil; U, United States).

not *Glossophaga* (trivial because it occurs on a single tip) or *Molossus*. The latter result is unexpected because, with the exception of spillovers, the *Molossus* clade is monophyletic (Fig. 2) and, hence, should have a strong phylogeny-trait association. This exception can be interpreted as a limitation of the method since the MC statistic has low resolution, reduced power, and incorrect type 1 error rates (10). The MC was significant for the three sampling countries (Argentina, the United States, and Brazil). Regarding clade B, the MC statistic was significant for all host genera (except for *Perimyotis*, occurring on a single tip) and for all countries except the United States (Table 1). This is a reasonable result since none of the five CoV sequences from the United States form a monophyletic clade.

The highest pairwise distance (PD) values observed were those of *Molossus* and *Eptesicus* in clade A and *Myotis* in clade B (Fig. 4 and 5; see also Table S3 in the supplemental material), with the latter being the highest value for both data sets.

We found significant and negative standardized effect size (SES) mean PD (MPD) values, indicating strong phylogenetic clustering, within all bat genera for both clades A and B (Fig. 4 and 5; Table S3). The only exception was *Eptesicus* in clade A, which depicts

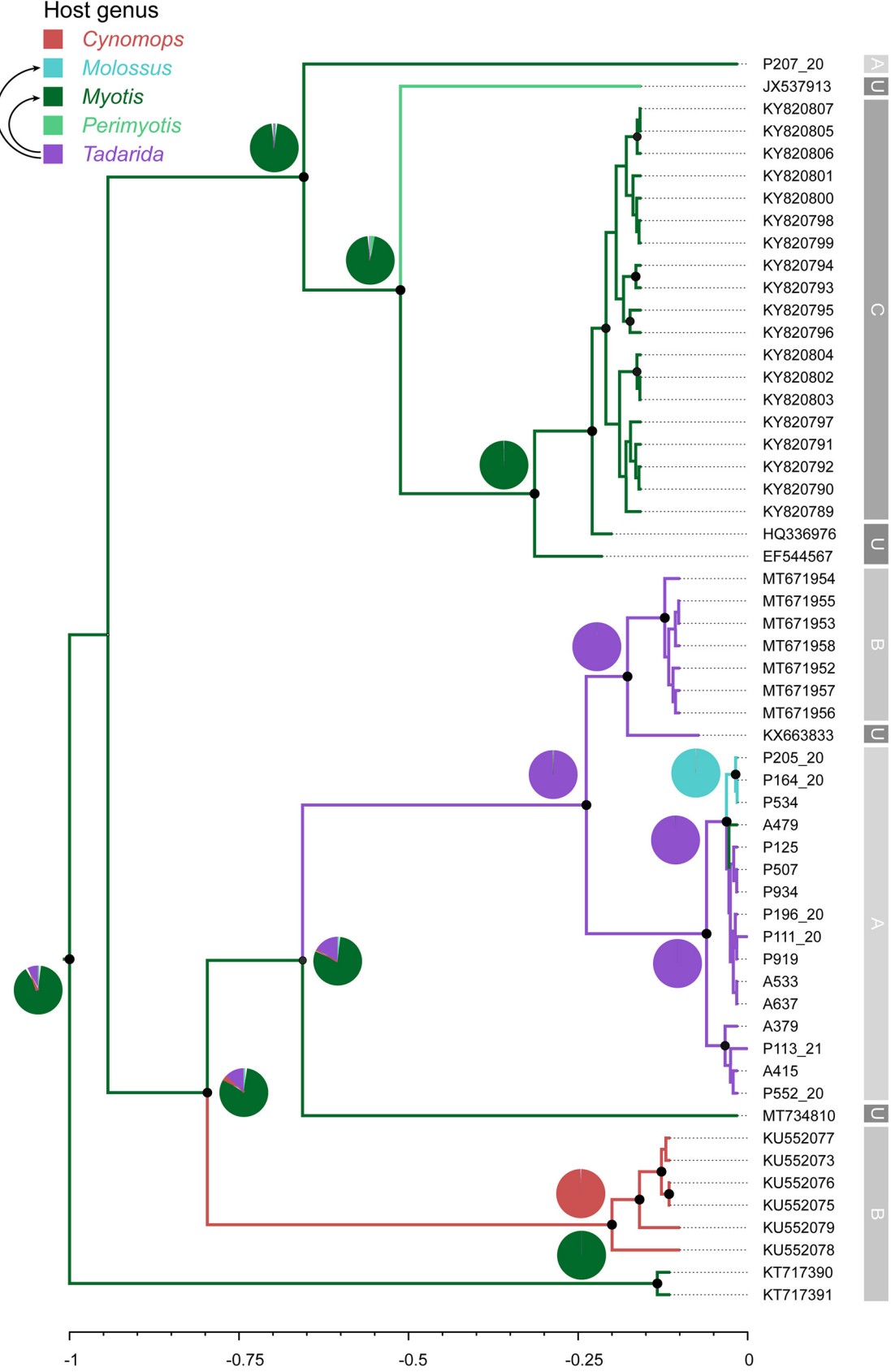

**FIG 3** Phylogenetic tree and ancestral host reconstruction of clade B. Maximum clade credibility trees were annotated using the alphaCoV data set of partial RdRp sequences and bat host genus as a discrete character state. Branch colors correspond to

**TABLE 1** Tip association statistics for hosts and geographic locations[a]

| | Value | | | | | | Significance (P value) |
|---|---|---|---|---|---|---|---|
| | Observed | | | Null | | | |
| Statistic | Mean | Lower 95% CI | Upper 95% CI | Mean | Lower 95% CI | Upper 95% CI | |
| **Clade A** | | | | | | | |
| Host | | | | | | | |
| AI | 0.488 | 0.086 | 0.850 | 1.982 | 1.494 | 2.422 | **<0.001** |
| PS | 4.755 | 4.000 | 5.000 | 12.307 | 10.653 | 13.617 | **<0.001** |
| MC (*Molossus*) | 3.785 | 3.000 | 6.000 | 2.906 | 2.078 | 4.786 | 0.453 |
| MC (*Eptesicus*) | 7.995 | 8.000 | 8.000 | 1.802 | 1.140 | 3.000 | **0.001** |
| MC (*Glossophaga*) | 1.000 | 1.000 | 1.000 | 1.000 | 1.000 | 1.000 | 1.000 |
| MC (*Tadarida*) | 3.000 | 3.000 | 3.000 | 1.054 | 1.000 | 1.332 | **0.001** |
| Location | | | | | | | |
| AI | 0.251 | 0.250 | 0.250 | 1.826 | 1.361 | 2.270 | **<0.001** |
| PS | 3.000 | 3.000 | 3.000 | 10.995 | 9.736 | 11.897 | **<0.001** |
| MC (Argentina) | 3.000 | 3.000 | 3.000 | 1.110 | 1.000 | 1.491 | **0.002** |
| MC (USA) | 7.995 | 8.000 | 8.000 | 1.531 | 1.004 | 2.063 | **0.001** |
| MC (Brazil) | 13.715 | 13.000 | 16.000 | 3.268 | 2.237 | 5.459 | **0.002** |
| **Clade B** | | | | | | | |
| Host | | | | | | | |
| AI | 0.179 | 0.005 | 0.384 | 3.878 | 3.248 | 4.472 | **<0.001** |
| PS | 5.022 | 5.000 | 5.000 | 24.888 | 22.403 | 27.212 | **<0.001** |
| MC (*Tadarida*) | 7.999 | 8.000 | 8.000 | 2.308 | 1.738 | 3.392 | **0.001** |
| MC (*Myotis*) | 21.003 | 21.000 | 21.000 | 2.960 | 2.112 | 4.086 | **0.001** |
| MC (*Perimyotis*) | 1.000 | 1.000 | 1.000 | 1.000 | 1.000 | 1.000 | 1.000 |
| MC (*Cynomops*) | 6.000 | 6.000 | 6.000 | 1.174 | 1.000 | 1.734 | **0.001** |
| MC (*Molossus*) | 2.950 | 3.000 | 3.000 | 1.036 | 1.000 | 1.243 | **0.001** |
| Location | | | | | | | |
| AI | 0.003 | 0.001 | 0.003 | 4.317 | 3.698 | 4.923 | **<0.001** |
| PS | 6.069 | 6.000 | 7.000 | 29.023 | 26.192 | 31.678 | **<0.001** |
| MC (Argentina) | 16.000 | 16.000 | 16.000 | 2.064 | 1.484 | 3.062 | **0.001** |
| MC (USA) | 1.000 | 1.000 | 1.000 | 1.112 | 1.000 | 1.481 | 1.000 |
| MC (Brazil) | 7.007 | 7.000 | 7.000 | 1.883 | 1.315 | 2.933 | **0.001** |
| MC (Canada) | 18.461 | 12.000 | 19.000 | 2.295 | 1.644 | 3.347 | **0.001** |

[a]Significance was assessed by comparing each statistic obtained by the posterior samples of each run against null distributions generated from 1,000 randomizations of traits to tips along each sampled tree. The significance level chosen was a P value of 0.05 (significant values are shown in boldface type). AI, association index; PS, parsimony score; MC, monophyletic clade size statistic; CI, confidence interval.

nonsignificant negative SES MPD values. Negative and significant SES mean nearest-taxon distance (MNTD) values were also observed for most genera in both clades, reflecting the phylogenetic structure closer to the tips. However, there are exceptions to this pattern. Still negative but nonsignificant SES MNTD values were observed in *Molossus* and *Eptesicus* (clade A) and *Cynomops* (clade B). The lower values observed for the first two can be attributed to spillovers that may alter the phylogenetic structure of the tips, even in the presence of strong basal clustering. The only positive SES MNTD value was that of *Myotis* (clade B). This is an expected result since *Myotis* hosts multiple sublineages in this clade (Fig. 3). The clusters based on the among-host phylogenetic distances reflect the phylogenetic relationships of the main host variants.

## DISCUSSION

The importance of evidence-based strategies for monitoring viral diversity in bat reservoir hosts has been revealed in an unprecedented way since the emergence of

**FIG 3** Legend (Continued)

the inferred ancestral bat genus with the highest posterior probability. Pie charts represent uncertainty in the ancestral host reconstruction. Black arrows depict events of cross-species transmission between bat genera. Branch lengths are scaled to relative time units. Relevant nodes with a posterior probability of >0.9 are indicated with black circles. Gray blocks at the right of the tree indicate the country (A, Argentina; B, Brazil; C, Canada; U, United States).

## Clade A

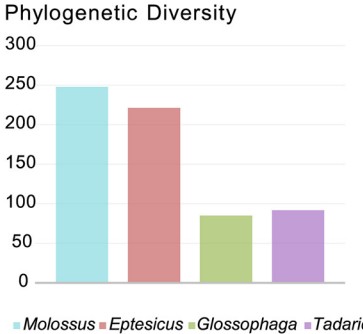

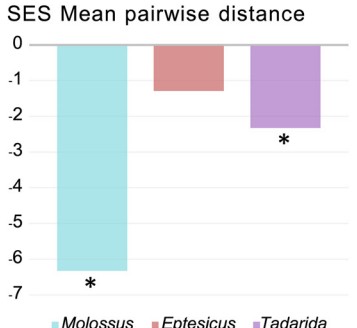

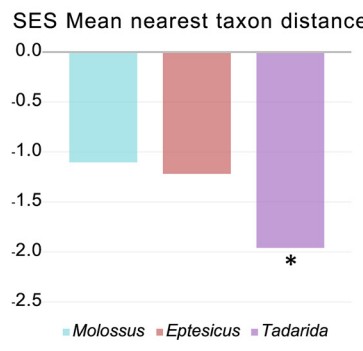

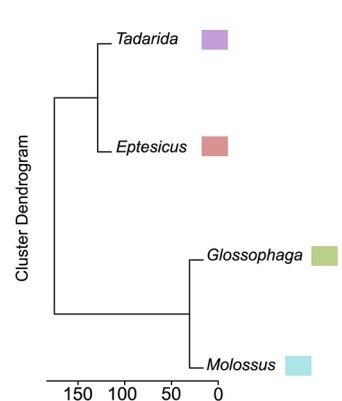

**FIG 4** Phylogenetic diversity analyses. Metrics of CoV phylogenetic diversity within each bat genus are shown for clade A. (Top left) Phylogenetic diversity; (top right) standardized effect size (SES) of the mean phylogenetic distance; (bottom left) standardized effect size of the mean nearest-taxon distance; (bottom right) phylogenetic ordination among bat host genera for alphaCoVs. One-tailed $P$ values (quantiles) were calculated after randomly reshuffling tip labels 1,000 times along the entire phylogeny. Values departing significantly from the null model ($P$ value of $<0.05$) are indicated with an asterisk. All exact $P$ values are available in Table S3 in the supplemental material.

the SARS-CoV-2 pandemic. Coronaviruses have been the subject of concern as potential pandemic threats, and there is still a challenge in specifying the ecological and evolutionary drivers of viral emergence. A recent global analysis showed that there is a clear bias in both geographic and taxonomic bat sampling, with a high preponderance for China compared to gaps throughout South Asia, the Americas, sub-Saharan Africa, and East Africa (11). In the Americas, the picture is far from complete; bat CoVs have been detected in 12 countries (including this study), with a bias toward specific regions, particularly Brazil, where additional bat surveillance work has recently taken place (9). Although there were recent advances in studies of phylogeography and host-shifting patterns in bat CoVs from this region (8), there is an obvious need to interpret the addition of novel sequences from unstudied species and geographic regions.

The diversity of bat CoVs in the Americas can be analyzed from two perspectives, according to virus and host heterogeneity. It is known that both alpha- and betaCoVs circulate in American bats, but there is a markedly uneven representation of these two genera. While betaCoVs have been found in a limited number of species, almost exclusively Phyllostomidae and Mormoopidae, there is a clear predominance of alphaCoVs (7, 8). American bat alphaCoVs can be divided into seven reciprocally monophyletic clades, four of which are exclusive to Phyllostomidae (clades D to F), while the remaining three (clades A to C) are maintained by members of the Vespertilionidae and Molossidae (Fig. 1). One of these clades (clade C) is closely related to clade B and is found exclusively in *Myotis*. The other two lineages, clades A and B, circulate in both bat families and are the focus of the present study.

Clades A and B show marked host and geographic structures, as shown by the tip association tests and the phylogenetic diversity analyses (Fig. 4 and 5 and Table 1; see also

## Clade B

### Phylogenetic Diversity

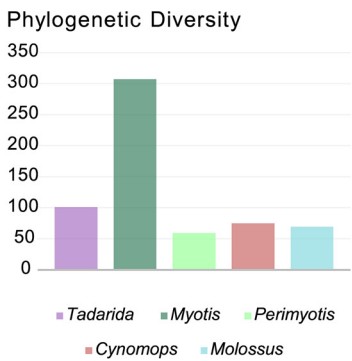

■ Tadarida ■ Myotis ■ Perimyotis
■ Cynomops ■ Molossus

### SES Mean pairwise distance

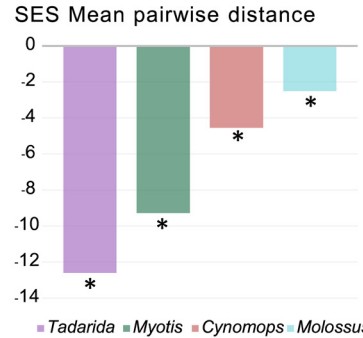

■ Tadarida ■ Myotis ■ Cynomops ■ Molossus

### SES Mean nearest taxon distance

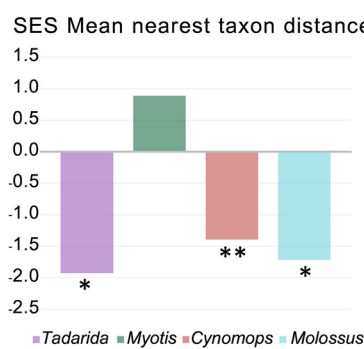

■ Tadarida ■ Myotis ■ Cynomops ■ Molossus

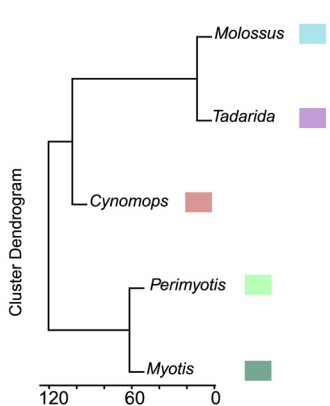

**FIG 5** Phylogenetic diversity analyses. Metrics of CoV phylogenetic diversity within each bat genus are shown for clade B. (Top left) Phylogenetic diversity; (top right) standardized effect size (SES) of the mean phylogenetic distance; (bottom left) standardized effect size of the mean nearest-taxon distance; (bottom right) phylogenetic ordination among bat host genera for alphaCoVs. One-tailed $P$ values (quantiles) were calculated after randomly reshuffling tip labels 1,000 times along the entire phylogeny. Values departing significantly from the null model ($P$ value of <0.05) are indicated with an asterisk. All exact $P$ values are available in Table S3 in the supplemental material.

Table S3 in the supplemental material). At the host level, there is also a basal structure, as revealed by the SES MPD analyses. Shallow host shifts took place in geographic regions that correspond to overlapping host ranges. Taken together, these results indicate that CoVs have high genus-specific tropism and are geographically restricted and that host switching reflects sympatry between bat species (Fig. 2 and 3).

Although both viral clades are maintained by vespertilionids and molossids, in most bat genera, only one of these two viral clades circulates (Fig. 1 to 3). However, there is an interesting exception. We found that two nonrelated lineages (clades A and B) cocirculate in *Tadarida* and *Molossus* in the same region (Argentina and Brazil) (Fig. 2 and 3). Notably, there are contrasting host-shifting scenarios that indicate different origins of these variants. In clade A, *Tadarida* and *Molossus* maintain nonrelated variants. The *Tadarida* sublineage would derive from *Eptesicus* (or an unsampled taxon) but is definitely not associated with *Molossus* (Fig. 2). In contrast, there is a clear relationship between these bat genera in clade B, supported by a recent host shift, where *Tadarida* is the donor and *Molossus* is the recipient host (Fig. 3). Overall, these results indicate that *Tadarida* and *Molossus* share different variants and are capable of maintaining high levels of CoV diversity in the same geographic region. *Tadarida* and *Molossus*, two synanthropic insectivorous bat genera with overlapping distributions, are prone to sharing mixed-species roosts. This association was previously confirmed by CoV sequence analysis of fecal samples from a maternity roost cohabited by *Molossus molossus* and *Tadarida brasiliensis* in Brazil (12). This observation is indirect evidence of the plausibility of the observed CST in clade B (Fig. 3).

*Myotis* is probably the major evolutionary reservoir of alphaCoVs in the Americas. As mentioned above, two of the three vespertilionid/molossid clades are maintained

totally (clade C) or partially (clade B) by *Myotis*, covering the complete geographic distribution of the studied region. Its widespread occurrence, together with being the richest host in terms of viral phylogenetic diversity, and that it has several representative lineages across the phylogeny, including basally splitting lineages, pose *Myotis* as the most likely ancestor for clades B and C (Fig. 1 and 3). The role of *Myotis* as a reservoir and a key link in the diversification and evolution of CoVs appears to be a global trend since, for example, *Myotis* has been involved in several significant host switches during alphaCoV evolution in China (13) and harbors different alphaCoV variants in America, Asia, Europe, and Oceania (8, 14–16). *Myotis* has been shown to maintain alphaCoV infection through hibernation as an apparently nonpathogenic infection due to low levels of physiological and physical activity and low levels of inflammation (17). The maintenance of CoV loads for long periods may also influence the evolvability of the virus, generating the conditions for the rise of new variants through mutation and recombination.

Although this study makes a relevant contribution to the knowledge of the diversity of alphaCoVs and their respective bat hosts in the Americas, there are some limitations that should be taken into consideration. First, our study is based on partial RNA-dependent RNA polymerase (RdRp) sequences. As occurs with the data set produced in this study, the majority of bat CoV sequences available in public databases were generated with the primers designed by Watanabe et al. (18), which produce a fragment of 440 to 800 bp. However, it should be noted that the RdRp gene reflects vertical ancestry and is less prone to recombination than other regions of the CoV genome (19). In addition, it is worth noting that the general structure of the viral phylogenies inferred in this study showed high levels of resolution with highly supported nodes. Another aspect that should be taken into consideration is the uneven geographic coverage of bat hosts and viruses. Clades A and B comprise samples from North America (the United States and Canada) and South America (Brazil and Argentina), and it is likely that unsampled species/lineages are to be discovered along the gap between these countries. For this reason, we did not focus on deep phylogenetic relationships and host shifts, and we specifically discussed those occurring at shallow levels of the phylogeny between sympatric taxa that reflect traceable CSTs.

In a previous study, it was proposed that Phyllostomidae played a key role in the long-term evolution of alphaCoVs in the Americas based on high CST rates and the maintenance of four distinct CoV clades (8). In this paper, we contribute a deeper insight into variants maintained by molossids and vespertilionids, which together have levels of diversity comparable to those of Phyllostomidae. These findings can be useful to conduct targeted discovery of bat CoVs, especially in *Myotis*, as well as to study in detail the cocirculation of two unrelated variants in *Molossus* and *Tadarida*. As mentioned above, the picture is far from complete, and more CoV sampling efforts are needed throughout the continent. Rare taxa may also function as hidden links between geographically separated sister clades. Another pending approach is the generation of whole-genome sequences to study recombination and selection related to host shifting. Coronavirus research in bats is still taking its first steps in the Americas, and more surveillance efforts and analytical approaches should be carried out to identify natural reservoirs and their patterns of cross-species transmission to have a holistic view of potential risks for both wildlife and the human population.

## MATERIALS AND METHODS

**Samples.** A total of 594 rectal tissue samples from insectivorous bats were obtained from 2019 to 2021 from several Argentinian provinces (see Table S1 in the supplemental material). The specimens were collected through rabies virus surveillance carried out by the Avellaneda Zoonosis Center, Buenos Aires province (BAI), and the Luis Pasteur Zoonosis Institute, Autonomous City of Buenos Aires (CABA).

**RNA extraction, PCR amplification, and sequencing.** Rectal tissue samples were suspended in a buffer containing guanidinium isothiocyanate in order to lyse the largest amount of tissue cells (20). The samples were allowed to stand for 15 min and then vortexed for 15 s and centrifuged for 10 min at 4°C at 5,000 rpm. From the supernatant obtained, viral RNA extraction was performed using a commercial kit based on magnetic beads, the MagMAX viral/pathogen nucleic acid isolation kit (catalog no. A42352; Thermo Fisher), using the automated Kingfisher equipment (Thermo Fisher), according to the manufacturer's instructions. In each processing plate, negative extraction controls were used by adding 140 $\mu$L of PCR-quality water, and positive extraction controls were used by adding 140 $\mu$L of a Mebus strain (diluted 10 to 4), corresponding to bovine coronavirus, provided by the National Institute of Agricultural Technology (INTA) (COD FR7-6/IR 302/17).

An initial screen for CoVs was performed using a real-time reverse transcription-quantitative PCR (RT-qPCR) assay (21). This multiplex assay, called pan-CoV RT-qPCR, is based on the use of three LNA (locked nucleic acid) probes, which are marked with different fluorophores and cover the variability of all known CoVs, allowing the provisional classification of the detected CoVs. A 179-bp fragment of the viral polymerase gene (RdRp) was amplified. Reactions were performed on an ABI 7500 instrument (Applied Biosystems) using the SuperScript III Platinum one-step qRT-PCR kit (catalog no. 11732-020; Invitrogen). All samples tested positive for the $\beta$-actin housekeeping gene, proving efficient RNA extraction.

The molecular characterization of the detected coronaviruses was carried out by means of a second heminested RT-PCR assay, which allowed the amplification of a fragment of the RdRp coding sequence of approximately 440 bp (18). When possible, amplicons that yielded high-quality PCR products of the initial amplicon (ca. 800 bp) were sequenced. Amplification products were purified using the ExoSAP-IT enzyme (USB Corporation), according to the manufacturer's instructions. Subsequently, the products were sequenced in both directions using the BigDye Terminator cycle sequencing reagent v3.1 (Applied Biosystems). Postsequencing purification was performed with the BigDye Xterminator commercial kit (Applied Biosystems), and the corresponding capillary electrophoresis was performed on the ABI3500 genetic analyzer automatic sequencing equipment (Hitachi, Applied Biosystems).

**Reference sequences.** A total of 594 CoV nucleotide sequences were retrieved from the Virus Pathogen Database and Analysis Resource (ViPR) (22) (http://www.viprbrc.org/), filtering for alphacoronavirus, bats, and the Americas in the genus, host, and location fields, respectively. A subset of 165 sequences encompassing the RdRp coding sequence was kept for the analysis. In addition, 44 recently published alphaCoV nucleotide sequences from Brazilian bat species were included (9). The sequence under GenBank accession no. KM514667 (*Tadarida brasiliensis*, Brazil) was removed because it was too short (202 bp) and had a null overlap in the majority of the sequences (Table S2).

**Global CoV phylogeny.** The sequences generated in this study as well as the American bat alphaCoV data set were tested for phylogenetic affinity using reference sequences for alpha-, beta-, gamma-, and deltaCoVs representing the global CoV diversity (23). Sequences were aligned with Clustal Omega (24). A Bayesian phylogeny was run using MrBayes 3.2.7 (25) through the CIPRES Gateway (26). The molecular substitution model was selected with Mr ModelTest v2 (27). A total of $5 \times 10^7$ Markov chain Monte Carlo (MCMC) generations were run under the general time reversible, plus a proportion of invariant sites and gamma-distributed rate heterogeneity among sites (GTR+I+G) model, with sampling every $5 \times 10^3$ generations, discarding the first 25% of the run (burn-in phase). The potential scale reduction factor (PSRF) and the average standard deviation of split frequencies (ASDSF) were used for convergence diagnostics. The "burn-in" phase fulfilled standard deviations of <0.01 and PSRF values of 1.00 to 1.02 for all estimated parameters. Trees were visualized in FigTree v1.4.4 (28). All tested sequences belonged to American alphaCoV clades A and B (8).

**Ancestral state reconstruction.** Host and geographic ancestral reconstructions of clades A and B were inferred using BEAST v1.10.4 software (29) through a Bayesian discrete phylogeographic approach. Sampling dates were used for molecular clock calibrations. The selected substitution models were GTR+G for both clades. Host, country, and date metadata associated with each sequence were retrieved using the rentrez package (30) in the R environment (31) with RStudio (32). All association tests and ancestral host reconstructions were based on bat genus and family. At the species level, there could be high rates of misidentification, especially in genera such as *Molossus* and *Myotis*, where there is an ongoing taxonomic revision, with several cryptic species that cannot be identified from external morphology and/or traditional single- or multilocus approaches (33).

A symmetric substitution model was selected for host ancestral state reconstruction. A strict clock and a constant population size coalescent prior were used for these analyses. Each run consisted of $1 \times 10^7$ MCMC generations (sample frequency, 1,000; burn-in, 10%). Convergence and mixing of the MCMC generations were analyzed using Tracer v1.7.2 (34), combining two independent runs with LogCombiner v1.10.4. Trees were summarized with the maximum clade credibility (MCC) option, discarding 10% of the run as the burn-in, using TreeAnnotator v1.10.4. We also ran the analysis with logNormal clocks and exponential growth population models, which yielded the same topology, and inconspicuous differences in branch lengths (data not shown). AquaPony software (35) was used to graphically represent the uncertainty in the ancestral host reconstruction.

**Association tests.** The clustering of traits among tips of clade A and clade B trees was tested using Bayesian Tip Association Significance Testing (BaTS) software (10). We assessed the clustering of bat genera and countries sampled. The significance of clustering for each trait was assessed by comparing the calculated association index (AI), parsimony score (PS), and monophyletic clade (MC) statistics for the posterior samples of each run (burn-in, 10%) against null distributions generated from 1,000 randomizations of traits to tips along each sampled tree. The significance level chosen was a $P$ value of 0.05.

**Phylogenetic diversity.** Different measures of phylogenetic diversity were taken to characterize bat CoVs from clades A and B using the R package picante (36). Phylogenetic diversity can be defined as the sum of the lengths of all of those branches on the tree that span the members of a set. The mean pairwise distance (MPD) and the mean nearest-taxon distance (MNTD) statistics and their standardized effect sizes (SESs) were also calculated for each bat genus. The MPD statistic measures the MPDs among all pairs of CoVs within a host, and therefore, it reflects phylogenetic structuring across the whole phylogenetic tree and assesses the overall divergence of CoV lineages in a community. The MNTD statistic measures the mean distance between each CoV and its nearest phylogenetic neighbor in a host, reflecting phylogenetic structuring closer to the tips (showing how locally clustered taxa are). SES MPD and SES MNTD values correspond to the differences between the phylogenetic distances in the observed communities and the null communities generated by randomizing tip labels. The SES values were calculated by randomly reshuffling tip labels 1,000 times along the entire phylogeny. Negative SES values and low quantiles denote phylogenetic clustering, and high quantiles and positive values indicate phylogenetic

evenness or greater phylogenetic distances among sequences from the same host genus, while values close to zero show a random dispersion.

The interhost values of MPD (equivalent to phylogenetic $\beta$-diversity), corresponding to the MPD among all pairs of CoVs from two distinct hosts, and their SESs were estimated using the comdist function of the R package phylocom (37).

**Data availability.** The nucleotide sequences generated in this study have been deposited in the GenBank sequence database (accession no. OP169150 to OP169170).

## SUPPLEMENTAL MATERIAL

Supplemental material is available online only.
**SUPPLEMENTAL FILE 1**, PDF file, 0.9 MB.

## ACKNOWLEDGMENTS

We thank Helena Lage Ferreira for providing nucleotide sequences of Brazilian alphaCoVs.

This work was funded by project no. 13 IP-COVID-19 no. 786 MinCyT and by the INEI-ANLIS Dr. Carlos G. Malbrán.

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
