## [Reviewer comments · Microbiology Spectrum]

Microbiology Spectrum

The role of Molossidae and Vespertilionidae in shaping the diversity of alphacoronaviruses in the Americas

Diego Caraballo, Maria Sabio, Valeria Colombo, Maria Piccirilli, Lorena Vico, Stella Maris Hirmas Riade, Josefina Campos, Gustavo Martinez, Fernando Beltran, Elsa Baumeister, and Daniel Cisterna

Corresponding Author(s): Diego Caraballo, Instituto de Ecología, Genética y Evolución de Buenos Aires

Review Timeline:

Submission Date:	August 10, 2022
Editorial Decision:	September 6, 2022
Revision Received:	September 9, 2022
Editorial Decision:	September 15, 2022
Revision Received:	September 16, 2022
Accepted:	September 20, 2022

Editor: Biao He

Reviewer(s): Disclosure of reviewer identity is with reference to reviewer comments included in decision letter(s). The following individuals involved in review of your submission have agreed to reveal their identity: Yousong Peng (Reviewer #2)

Transaction Report:

DOI: <https://doi.org/10.1128/spectrum.03143-22>

September 6, 2022

Dr. Diego A. Caraballo
Instituto de Ecología, Genética y Evolución de Buenos Aires
Universidad de Buenos Aires, CONICET
Intendente Guiraldes 2160
Ciudad Universitaria, Pabellón 2, 4to piso
Buenos Aires 1428
Argentina

Re: Spectrum03143-22 (**The role of Molossidae and Vespertilionidae in shaping the diversity of alphacoronaviruses in the Americas**)

Dear Dr. Diego A. Caraballo:

Thank you for submitting your manuscript to Microbiology Spectrum.

Bats are among the natural hosts of coronaviruses (CoVs) and a large number of bat-CoVs have been reported worldwide. However, the diversity of bat-CoVs is currently underrepresented in the Americas, so your data contributes to knowledge of the genetic diversity and circulation of bat-CoVs in the area. The sequences in this study, as noted by the reviewer(s), are very short in length and have a small population size, which means they cannot adequately support some of the bioinformatic conclusions. I would suggest you emphasize the detection and genetic diversity of bat-CoVs and weaken the Bayesian inference, otherwise add more data to support the analysis.

Link Not Available

Sincerely,

Biao He

Journals Department
Reviewer comments:

Reviewer #1 (Comments for the Author):

Caraballo et al reported alpha-covs discovered in Molossidae and Vespertilionidae bats. The design of the study is similar to other bat cov studies, without too much novelty. The cross-species of the and diversity of covs and shaping ability of bats are discussed everywhere. Besides, the authors used a partial gene to do the analyses, which greatly reduced the credibility of the designed research. Also, the numbers of species was too small to make the bayesian inference less credible.

Reviewer #2 (Comments for the Author):

The study analyzed the role of two bat families, i.e., Molossidae and Vespertilionidae, in shaping the diversity of alphaCoVs in the Americas based on phylogenetic analysis. The study is interesting and helps deepen our understanding about the evolution of alphaCoVs in America. The manuscript was well written and clearly organized. The review has some minor concerns.

1 The authors used the strict clock model and a constant population size coalescent prior in Bayesian analysis. How about other clock models and population models?

2 Suggests to add some sub-headings in the Results section, which would make the Results section more user-friendly to the reader.

3 Authors used a segment of 440 nt of RdRP for analysis in the study. Please specify the detailed location of this segment.

4 In Figure 2 and 3, suggests to add arrows to the spillover events in the tree.

5 In the Introduction section, the genus name of coronavirus should be italic.

6 "Bat-CoVs, are likely the ancestral hosts for all alpha- and beta- human-infective coronaviruses", CoVs are not hosts.

7 In the Method section, the terms of "5E7", "5E3", and so on, are not standard in scientific writing.

Staff Comments:

Preparing Revision Guidelines

Please return the manuscript within 60 days; if you cannot complete the modification within this time period, please contact me. If you do not wish to modify the manuscript and prefer to submit it to another journal, please notify me of your decision immediately so that the manuscript may be formally withdrawn from consideration by Microbiology Spectrum.

Reviewer comments

Reviewer #1 (Comments for the Author):

Caraballo et al reported alpha-covs discovered in Molossidae and Vespertilionidae bats. The design of the study is similar to other bat cov studies, without too much novelty. The cross-species of the and diversity of covs and shaping ability of bats are discussed everywhere. Besides, the authors used a partial gene to do the analyses, which greatly reduced the credibility of the designed research. Also, the numbers of species was too small to make the bayesian inference less credible.

Response:

We have a partial agreement with the Reviewer. The cross-species transmission of bat-CoVs in the Americas was not studied until a paper recently published by one of us (Caraballo 2022). In that paper, the clades that are the focus of this study were underrepresented, basically because there is a significant proportion of sequences generated in the present study. However, as we state below, we decided to narrow the CST analysis, in agreement with some weak points identified by the Reviewer.

We acknowledge that there is an evident gap in the geographical coverage of bat hosts and viruses. This was stated in the previous version of the manuscript. We agree to narrow the scope of our analysis and the conclusions drawn from it, but we would like to mention that our study contributes valuable results that enable gaining insight into the diversity of alphaCoVs in the Americas. To name some, we found independent lineages co-circulating in *Tadarida* and *Molossus*, as well as identified a host shift between these bat taxa. We also made a thorough analysis of the phylogenetic structure and diversity of the clades maintained by vespertilionids and molossids, showing that there is a strong host and geographic structure. Finally, we highlight *Myotis* as the main reservoir in terms of phylogenetic diversity. These conclusions were reached independently of the Bayesian inference of host shifts and the analysis of basal relationships among lineages (that could be influenced by uneven geographical sampling), which were excluded from the study after the Reviewer's comments.

As, except for host shifts, all main sub-lineages are reciprocally monophyletic, we centered our analysis on shallow CSTs, and not on their basal relationships. Uneven

geographic sampling may interfere with deep relationships between lineages, so we excluded the Bayes Factors inference (that accounts for both deep and shallow CSTs), and specifically discussed CSTs occurring at shallow levels of the phylogeny, between sympatric taxa which reflect traceable host shifts. In other words, it is likely that there would be missing links between two different bat genera occurring one in South America and the other one in North America. On the other hand, the host shift *Tadarida*->*Molossus* is likely, since the involved sequences were isolated from bats occurring in the same geographic region. The same applies to the remaining host shifts treated in the manuscript.

As to the number of sequences obtained, we would like to clarify that the aim of this work was not to perform a population study, but a phylogenetic one. We do not use sequences to infer demographic parameters, and even when molecular clocks are used, we do not treat time in absolute units. So, all our analysis derives from topology and branch lengths, which are essentially phylogenetic parameters, that would not require a denser population sampling.

We also acknowledge that there are limitations arising from the use of partial RNA-dependent RNA polymerase (RdRp) sequences. As occurs with the dataset produced in this study, the majority of bat-CoV sequences available in public databases were generated with the primers designed by Watanabe et al. (2010) which produce a fragment of 440 bp-800. However, it should be noted that the RdRp gene reflects vertical ancestry and is less prone to recombination than other regions of the CoV genome. In addition, it is worth noting that the general structure of the viral phylogenies inferred in this study showed high levels of resolution with highly supported nodes.

The study was based on less than 200 RdRp sequences because there were no additional bat-CoV sequences for the region studied (the Americas). The procedure for selecting the final dataset was made by filtering for Bat coronavirus in the Americas through the VIPR website (accessed on 3rd June 2022). This search yielded 195 sequences, most of which were partial RdRp sequences (of 400 bp) amplified with a widespread primer set (Watanabe et al. 2010). A fraction of these sequences were further discarded because they spanned a non-overlapping region, different from the rest of the sequences. The final dataset consisted of 145 overlapping bat-CoV sequences, maximizing the breadth of the analysis.

To illustrate the enormous disparity between full genomes and partial batCoV sequences, we would like to share with you the following figure, which summarizes the number of available batCoV sequences of the studied lineages.

The vast majority of available sequences are short (<500 bp). There are only 6 nearly full genomes (clade A: all of them *Eptesicus*, clade B: no full genomes) which are insufficient for any analysis. The majority of the sequences produced in this study are over the median length of available sequences (426 bp).

We agree that this dataset has inherent limitations, and as such, the conclusions that follow this analysis must be taken with precaution (and this is reflected in the Discussion section). But if a strict criterion of sequence length is applied, the analysis cannot be done, and the opportunity to start gaining insight into batCoVs maintained by vespertilionids and molossids in the Americas would be lost. We are confident that there will be more batCoV sequences available in the next few years (we are personally working on generating full genome sequences in Argentina), but we are also convinced that these results could serve as good working hypotheses to guide future research and to point out precisely how important it would be to count with genomic analyses of virus and hosts diversity in the Americas.

Reviewer #2 (Comments for the Author):

The study analyzed the role of two bat families, i.e., Molossidae and Vespertilionidae, in shaping the diversity of alphaCoVs in the Americas based on phylogenetic analysis. The study is interesting and helps deepen our understanding about the evolution of alphaCoVs in America. The manuscript was well written and clearly organized. The review has some minor concerns.

1 The authors used the strict clock model and a constant population size coalescent prior in Bayesian analysis. How about other clock models and population models?

Response: We ran alternative models (logNormal clocks, and exponential growth) but these yielded the same topology and there were no conspicuous differences in branch lengths. As the aim of the study was not to infer demographic/temporal parameters, we decided to mention these results as “data not shown”, in order to avoid overextension of the manuscript and figures.

2 Suggests to add some sub-headings in the Results section, which would make the Results section more user-friendly to the reader.

Response: We took the Reviewer’s recommendation adding subheadings to the Results section.

3 Authors used a segment of 440 nt of RdRP for analysis in the study. Please specify the detailed location of this segment.

Response: We specified the location of the amplicon in the RdRp coding sequence, and relative to the ORF1ab.

4 In Figure 2 and 3, suggests to add arrows to the spillover events in the tree.

Response: As we restricted the analysis to shallow CSTs, we indicated the occurrence and direction of spillovers/host shifts in Figures 2 and 3, next to the colored boxes illustrating each bat taxon.

5 In the Introduction section, the genus name of coronavirus should be italic.

Response: We thank the Reviewer for the close reading. We italicized the genus names.

6 "Bat-CoVs, are likely the ancestral hosts for all alpha- and beta- human-infective coronaviruses", CoVs are not hosts.

Response: We thank the Reviewer for the close reading. We fixed that error.

7 In the Method section, the terms of "5E7", "5E3", and so on, are not standard in scientific writing.

Response: We replaced these expressions with "5x10⁷", "5x10³", etc.

September 15, 2022

Dr. Diego A. Caraballo
Instituto de Ecología, Genética y Evolución de Buenos Aires
Universidad de Buenos Aires, CONICET
Intendente Guiraldes 2160
Ciudad Universitaria, Pabellón 2, 4to piso
Buenos Aires 1428
Argentina

Re: Spectrum03143-22R1 (**The role of Molossidae and Vespertilionidae in shaping the diversity of alphacoronaviruses in the Americas**)

Dear Dr. Diego A. Caraballo:

Please make sure these sequences reported in the study are publicly available, and you should provide a "Data Availability" paragraph at the end of the Materials and Methods section.

Thank you for submitting your manuscript to Microbiology Spectrum. As you will see your paper is very close to acceptance. Please modify the manuscript along the lines I have recommended. As these revisions are quite minor, I expect that you should be able to turn in the revised paper in less than 30 days, if not sooner. If your manuscript was reviewed, you will find the reviewers' comments below.

When submitting the revised version of your paper, please provide (1) point-by-point responses to the issues raised by the reviewers as file type "Response to Reviewers," not in your cover letter, and (2) a PDF file that indicates the changes from the original submission (by highlighting or underlining the changes) as file type "Marked Up Manuscript - For Review Only". Please use this link to submit your revised manuscript. Detailed instructions on submitting your revised paper are below.

Link Not Available

Sincerely,

Biao He

Reviewer comments:

Preparing Revision Guidelines

- Point-by-point responses to the issues raised by the reviewers in a file named "Response to Reviewers," NOT IN YOUR COVER LETTER.
- Upload a compare copy of the manuscript (without figures) as a "Marked-Up Manuscript" file.
- Each figure must be uploaded as a separate file, and any multipanel figures must be assembled into one file.
- Manuscript: A .DOC version of the revised manuscript

- Figures: Editable, high-resolution, individual figure files are required at revision, TIFF or EPS files are preferred

Please return the manuscript within 60 days; if you cannot complete the modification within this time period, please contact me. If you do not wish to modify the manuscript and prefer to submit it to another journal, please notify me of your decision immediately so that the manuscript may be formally withdrawn from consideration by Microbiology Spectrum.

Reviewer comments

Reviewer #1 (Comments for the Author):

Caraballo et al reported alpha-covs discovered in Molossidae and Vespertilionidae bats. The design of the study is similar to other bat cov studies, without too much novelty. The cross-species of the and diversity of covs and shaping ability of bats are discussed everywhere. Besides, the authors used a partial gene to do the analyses, which greatly reduced the credibility of the designed research. Also, the numbers of species was too small to make the bayesian inference less credible.

Response:

We have a partial agreement with the Reviewer. The cross-species transmission of bat-CoVs in the Americas was not studied until a paper recently published by one of us (Caraballo 2022). In that paper, the clades that are the focus of this study were underrepresented, basically because there is a significant proportion of sequences generated in the present study. However, as we state below, we decided to narrow the CST analysis, in agreement with some weak points identified by the Reviewer.

We acknowledge that there is an evident gap in the geographical coverage of bat hosts and viruses. This was stated in the previous version of the manuscript. We agree to narrow the scope of our analysis and the conclusions drawn from it, but we would like to mention that our study contributes valuable results that enable gaining insight into the diversity of alphaCoVs in the Americas. To name some, we found independent lineages co-circulating in *Tadarida* and *Molossus*, as well as identified a host shift between these bat taxa. We also made a thorough analysis of the phylogenetic structure and diversity of the clades maintained by vespertilionids and molossids, showing that there is a strong host and geographic structure. Finally, we highlight *Myotis* as the main reservoir in terms of phylogenetic diversity. These conclusions were reached independently of the Bayesian inference of host shifts and the analysis of basal relationships among lineages (that could be influenced by uneven geographical sampling), which were excluded from the study after the Reviewer's comments.

As, except for host shifts, all main sub-lineages are reciprocally monophyletic, we centered our analysis on shallow CSTs, and not on their basal relationships. Uneven

geographic sampling may interfere with deep relationships between lineages, so we excluded the Bayes Factors inference (that accounts for both deep and shallow CSTs), and specifically discussed CSTs occurring at shallow levels of the phylogeny, between sympatric taxa which reflect traceable host shifts. In other words, it is likely that there would be missing links between two different bat genera occurring one in South America and the other one in North America. On the other hand, the host shift *Tadarida*->*Molossus* is likely, since the involved sequences were isolated from bats occurring in the same geographic region. The same applies to the remaining host shifts treated in the manuscript.

As to the number of sequences obtained, we would like to clarify that the aim of this work was not to perform a population study, but a phylogenetic one. We do not use sequences to infer demographic parameters, and even when molecular clocks are used, we do not treat time in absolute units. So, all our analysis derives from topology and branch lengths, which are essentially phylogenetic parameters, that would not require a denser population sampling.

We also acknowledge that there are limitations arising from the use of partial RNA-dependent RNA polymerase (RdRp) sequences. As occurs with the dataset produced in this study, the majority of bat-CoV sequences available in public databases were generated with the primers designed by Watanabe et al. (2010) which produce a fragment of 440 bp-800. However, it should be noted that the RdRp gene reflects vertical ancestry and is less prone to recombination than other regions of the CoV genome. In addition, it is worth noting that the general structure of the viral phylogenies inferred in this study showed high levels of resolution with highly supported nodes.

The study was based on less than 200 RdRp sequences because there were no additional bat-CoV sequences for the region studied (the Americas). The procedure for selecting the final dataset was made by filtering for Bat coronavirus in the Americas through the VIPR website (accessed on 3rd June 2022). This search yielded 195 sequences, most of which were partial RdRp sequences (of 400 bp) amplified with a widespread primer set (Watanabe et al. 2010). A fraction of these sequences were further discarded because they spanned a non-overlapping region, different from the rest of the sequences. The final dataset consisted of 145 overlapping bat-CoV sequences, maximizing the breadth of the analysis.

To illustrate the enormous disparity between full genomes and partial batCoV sequences, we would like to share with you the following figure, which summarizes the number of available batCoV sequences of the studied lineages.

The vast majority of available sequences are short (<500 bp). There are only 6 nearly full genomes (clade A: all of them *Eptesicus*, clade B: no full genomes) which are insufficient for any analysis. The majority of the sequences produced in this study are over the median length of available sequences (426 bp).

We agree that this dataset has inherent limitations, and as such, the conclusions that follow this analysis must be taken with precaution (and this is reflected in the Discussion section). But if a strict criterion of sequence length is applied, the analysis cannot be done, and the opportunity to start gaining insight into batCoVs maintained by vespertilionids and molossids in the Americas would be lost. We are confident that there will be more batCoV sequences available in the next few years (we are personally working on generating full genome sequences in Argentina), but we are also convinced that these results could serve as good working hypotheses to guide future research and to point out precisely how important it would be to count with genomic analyses of virus and hosts diversity in the Americas.

Reviewer #2 (Comments for the Author):

The study analyzed the role of two bat families, i.e., Molossidae and Vespertilionidae, in shaping the diversity of alphaCoVs in the Americas based on phylogenetic analysis. The study is interesting and helps deepen our understanding about the evolution of alphaCoVs in America. The manuscript was well written and clearly organized. The review has some minor concerns.

1 The authors used the strict clock model and a constant population size coalescent prior in Bayesian analysis. How about other clock models and population models?

Response: We ran alternative models (logNormal clocks, and exponential growth) but these yielded the same topology and there were no conspicuous differences in branch lengths. As the aim of the study was not to infer demographic/temporal parameters, we decided to mention these results as “data not shown”, in order to avoid overextension of the manuscript and figures.

2 Suggests to add some sub-headings in the Results section, which would make the Results section more user-friendly to the reader.

Response: We took the Reviewer’s recommendation adding subheadings to the Results section.

3 Authors used a segment of 440 nt of RdRP for analysis in the study. Please specify the detailed location of this segment.

Response: We specified the location of the amplicon in the RdRp coding sequence, and relative to the ORF1ab.

4 In Figure 2 and 3, suggests to add arrows to the spillover events in the tree.

Response: As we restricted the analysis to shallow CSTs, we indicated the occurrence and direction of spillovers/host shifts in Figures 2 and 3, next to the colored boxes illustrating each bat taxon.

5 In the Introduction section, the genus name of coronavirus should be italic.

Response: We thank the Reviewer for the close reading. We italicized the genus names.

6 "Bat-CoVs, are likely the ancestral hosts for all alpha- and beta- human-infective coronaviruses", CoVs are not hosts.

Response: We thank the Reviewer for the close reading. We fixed that error.

7 In the Method section, the terms of "5E7", "5E3", and so on, are not standard in scientific writing.

Response: We replaced these expressions with "5x10⁷", "5x10³", etc.

September 20, 2022

Dr. Diego A. Caraballo
Instituto de Ecología, Genética y Evolución de Buenos Aires
Universidad de Buenos Aires, CONICET
Intendente Guiraldes 2160
Ciudad Universitaria, Pabellón 2, 4to piso
Buenos Aires 1428
Argentina

Re: Spectrum03143-22R2 (**The role of Molossidae and Vespertilionidae in shaping the diversity of alphacoronaviruses in the Americas**)

Dear Dr. Diego A. Caraballo:

I am glad to inform you that your manuscript has been accepted, and I am forwarding it to the ASM Journals Department for publication. You will be notified when your proofs are ready to be viewed.

Sincerely,

Biao He
Editor, Microbiology Spectrum
